# Old Therapy, New Questions: Rethinking Phlebotomy in a Pharmacologic Landscape

**DOI:** 10.3390/ph18081212

**Published:** 2025-08-16

**Authors:** Andrea Duminuco, Patrick Harrington, Vittorio Del Fabro, Elvira Scalisi, Gabriella Santuccio, Annalisa Santisi, Arianna Sbriglione, Bruno Garibaldi, Uros Markovic, Francesco Di Raimondo, Giuseppe Alberto Palumbo, Novella Pugliese, Calogero Vetro

**Affiliations:** 1Hematology Unit with BMT, A.O.U. Policlinico “G. Rodolico-San Marco”, 95123 Catania, Italy; andrea.elvira.scalisi@gmail.com (E.S.); santuccio.gabriella@gmail.com (G.S.); annalisa_santisi@hotmail.it (A.S.); sbriglionearianna@gmail.com (A.S.); brunga93@gmail.com (B.G.); urosmarkovic09041989@gmail.com (U.M.); francesco.diraimondo@unict.it (F.D.R.); 2Department of Haematology, Guy’s and St Thomas’ NHS Foundation Trust, London SE1 9RT, UK; patrick.harrington3@nhs.net; 3Faculty of Medicine and Surgery, “Kore” University of Enna, 94100 Enna, Italy; vittorio.delfabro@unikore.it; 4Department of Medical, Surgical Sciences and Advanced Technologies “G.F. Ingrassia”, University of Catania, 95123 Catania, Italy; palumbo.ga@gmail.com; 5Department of Clinical Medicine and Surgery, Hematology Section, University of Naples “Federico II”, Via Sergio Pansini, 5, 80131 Naples, Italy; novypugliese@yahoo.it; 6Hematology and Bone Marrow Transplantation Unit (BMTU), Hospital of Bolzano (SABES-ASDAA), Teaching Hospital of Paracelsus Medical University (PMU), 39100 Bolzano, Italy; gerovetro@gmail.com

**Keywords:** therapeutic phlebotomy, polycythemia vera, hematocrit control, targeted therapies

## Abstract

Therapeutic phlebotomy remains a key intervention in the management of erythrocytosis and iron overload disorders, particularly polycythemia vera (PV) and hereditary hemochromatosis. Despite its historical origins as an ancient practice, venesection continues to be recommended in international guidelines for the reduction of hematocrit and iron burden, thereby mitigating thrombotic and organ-related complications. However, the evolving landscape of targeted pharmacologic therapies is reshaping the therapeutic paradigm. This review examines the current role of therapeutic phlebotomy, with a particular focus on PV, outlining its physiological rationale, clinical benefits, and well-documented limitations—including iron deficiency, procedural burden, and incomplete hematocrit control between sessions. Comparative insights are provided between phlebotomy and red cell apheresis, highlighting differences in efficacy, tolerability, and accessibility. The emergence of disease-modifying agents—such as interferons, JAK inhibitors, hepcidin mimetics, and epigenetic modulators like givinostat and bomedemstat—promises more sustained hematologic control with the potential to reduce or eliminate the need for repeated phlebotomies. While phlebotomy remains indispensable in early-stage or low-risk PV, its future utility will likely shift toward complementary or bridge therapy in the context of individualized, pharmacologically driven strategies, redefining the role of phlebotomy in the era of precision medicine.

## 1. Introduction

Therapeutic phlebotomy, also known as venesection, represents one of the oldest medical procedures in recorded history, yet it remains a cornerstone of contemporary management for select hematological and iron overload disorders [1]. Historically rooted in humoral theories dating back to ancient Egypt and classical Greece, phlebotomy was widely practiced across civilizations, from Hippocrates and Galen to Islamic medicine and medieval Europe, as a therapeutic approach for conditions perceived to arise from an excess of blood or internal imbalance [2]. Although largely abandoned with the rise of evidence-based medicine in the early 20th century, its clinical utility re-emerged with the recognition of its efficacy in specific disease contexts.

In modern medicine, therapeutic phlebotomy is indicated primarily in three conditions: hereditary hemochromatosis, polycythemia vera (PV), and porphyria cutanea tarda. The procedure entails the scheduled withdrawal of whole blood, typically 450 to 500 mL per session, from a peripheral vein, with or without isovolemic fluid replacement depending on the patient’s tolerance and baseline cardiovascular status [3]. The goal is to reduce total blood volume or iron burden, or both, thereby mitigating pathological consequences such as hyperviscosity, thrombosis, or organ damage.

Despite its procedural simplicity, phlebotomy has profound systemic effects. In PV, it is employed to reduce hematocrit and lower thrombotic risk, while in hemochromatosis, it serves to deplete excess iron stores, preventing hepatic, cardiac, and endocrine complications [4]. Nonetheless, the intervention is not devoid of limitations: repetitive sessions may induce or exacerbate iron deficiency, fatigue, and reduced quality of life [5].

Given the increasing understanding of disease-specific pathophysiology and the advent of targeted therapies, the role of phlebotomy has been re-evaluated over the years. While still considered first-line therapy in many contexts, especially in early-stage PV or asymptomatic hemochromatosis, its utility must be weighed against patient burden, adverse effects, and the availability of novel pharmacologic alternatives, such as interferons or hepcidin mimetics [6].

This review aims to explore the multifaceted clinical role of therapeutic phlebotomy, from its historical origins to modern-day indications, with a focused lens on PV and, overall, myeloproliferative neoplasms (MPN), discussing the physiological rationale, therapeutic benefits and limitations, the evolving treatment paradigms, and the potential repositioning of phlebotomy in light of emerging disease-modifying therapies.

## 2. Physiology, Benefits and Adverse Effects of Phlebotomy

Therapeutic phlebotomy reduces red blood cell (RBC) mass and total body iron content, which in turn decreases blood viscosity and mitigates hyperviscosity-related complications [7]. These effects are central to the management of erythrocytosis in MPNs, particularly PV, and in conditions of iron overload such as hereditary hemochromatosis [8,9].

Beyond its mechanical effects, phlebotomy has profound physiological consequences, particularly in relation to iron metabolism, erythropoiesis, and cardiovascular risk.

### 2.1. Hemorheological Rationale and Viscosity Reduction

Elevated hematocrit (Hct) directly increases blood viscosity, impairing laminar flow and promoting turbulent shear stress on the vascular endothelium [10].

This activates endothelial cells, platelets, and leukocytes, fostering a pro-thrombotic environment [11,12,13]. In PV, these effects are exaggerated due to the clonal myeloid hyperproliferation driven by JAK2 mutations [14,15,16]. The CYTO-PV study provided the most robust evidence to date that maintaining Hct below 45% reduces the incidence of cardiovascular death and major thrombosis in PV [17]. As discussed below, this has since become the cornerstone of clinical management in PV and is widely endorsed in international guidelines.

### 2.2. Iron Metabolism: Stores, Toxicity, and Depletion

Iron plays a fundamental role in erythropoiesis, acting as a cofactor in hemoglobin synthesis [18]. In hemochromatosis, systemic iron overload results in its deposition as hemosiderin and ferritin in parenchymal tissues (primarily the liver, pancreas, heart, and joints) where it can cause oxidative damage through Fenton chemistry [19,20]. Phlebotomy effectively depletes circulating iron and prevents further organ injury by forcing mobilization from tissue stores. This mechanism underpins its role as first-line therapy in hereditary hemochromatosis [21], becoming a potential target in potentially incurable diseases [22].

Conversely, in PV and other erythrocytosis syndromes, repeated phlebotomy induces iron deficiency. Although this effect is somehow desired to limit erythropoiesis, it can lead to symptoms such as fatigue, poor concentration, and restless leg syndrome [23]. Iron depletion also impacts non-hematopoietic tissues, possibly contributing to systemic symptoms and impaired quality of life [24].

### 2.3. The Hepcidin–Ferroportin Axis

The regulation of systemic iron balance is centrally controlled by hepcidin, a 25-amino-acid peptide hormone synthesized by the liver. Hepcidin binds to and degrades ferroportin, a cellular iron exporter, thereby reducing dietary iron absorption and iron release from macrophages and hepatocytes [25,26]. In hereditary hemochromatosis, mutations in genes such as HFE, HJV, TFR2, and HAMP result in inappropriately low hepcidin levels, leading to uncontrolled iron absorption and subsequent systemic overload (Figure 1) [27].

Phlebotomy indirectly stimulates hepcidin production by lowering systemic iron stores and increasing erythropoietic drive, which induces feedback mechanisms involving erythroferrone and erythropoietin [28,29]. However, in PV, the erythropoietic signal often overrides iron-restrictive signals, resulting in continued erythropoiesis despite iron depletion, a phenomenon contributing to “masked PV” or Hct levels that seem normal despite high red cell mass [30].

### 2.4. Hepcidin Mimetics: Toward “Chemical Phlebotomy”

The advent of hepcidin mimetics such as rusfertide (PTG-300, see below) has opened new therapeutic avenues. These agents pharmacologically mimic hepcidin’s activity by blocking ferroportin-mediated iron export [31,32].

This “chemical phlebotomy” can offer the potential for more stable Hct control, improved quality of life, and fewer complications from iron depletion. Longitudinal and prospective studies, discussed in depth in the further chapters, are ongoing to determine the long-term impact of these agents on thrombotic risk and disease progression.

### 2.5. Other Adverse Effects of Phlebotomy

In addition to iron deficiency, phlebotomy may be associated with other side effects. It is not uncommon for individuals to experience discomfort when exposed to the sight of blood or needles. This can trigger a vasovagal response, a physiological reaction initiated by the nervous system. Symptoms may include dizziness, sweating, a decrease in heart rate or blood pressure, and, in some cases, fainting.

To mitigate these effects, relaxation techniques such as deep breathing may be recommended. Additionally, redirecting the individual’s attention to another object can serve as a helpful distraction.

Following the procedure, patients may experience temporary redness, bruising, or mild soreness at the puncture site. These side effects generally resolve shortly after the procedure [33,34].

In a retrospective study, between August 2021 and September 2022, a total of 587 phlebotomy procedures on 189 patients were analyzed. During the study period, 93% of patients underwent two or more procedures. Adverse events were reported in 20 patients (10.8%), accounting for 25 events overall (4.3% of phlebotomies). These were predominantly vasovagal reactions, none of which were clinically significant. All events were promptly managed on-site by nursing staff, with full recovery observed in all cases, confirming that even elderly patients and those with some comorbidities can safely undergo phlebotomy when the process is carefully managed [35].

Healthcare providers may advise patients to maintain adequate hydration, avoid alcohol, and refrain from vigorous physical activity for several hours post-procedure to promote recovery and minimize further complications.

Thrombocytosis was also described after phlebotomy. The hypothesis that post-phlebotomy thrombocytosis elevates thrombotic risk is primarily derived from the PVSG-01 study [36]. A separate observational study identified a subgroup of patients who required more frequent phlebotomies and demonstrated a higher incidence of thrombotic events [37], although without a significant rise in platelet counts. Furthermore, analyzing data from the ECLAP and CYTO-PV trials, it was found that the number of phlebotomies performed in patients treated with hydroxyurea did not correlate with an increased risk of thrombosis [38].

### 2.6. Differences Between Phlebotomy and Apheresis

Red blood cell (RBC) apheresis is a procedure involving the selective removal of erythrocytes using an automated cell separator, allowing for a more controlled and efficient reduction in hematocrit compared to conventional phlebotomy. A single session is often sufficient to bring hematocrit within the desired target range, offering a rapid and precise intervention, particularly beneficial in urgent clinical scenarios [39].

However, RBC-apheresis is not without drawbacks. Adverse events may include dizziness, paresthesia, muscle twitching, cramps, tetany, vasovagal syncope, fever, arrhythmias, and, in some cases, an exacerbation of thrombotic symptoms. These complications are generally transient and manageable but require appropriate monitoring during the procedure. The most significant limitation remains its high cost, which restricts its routine use in many healthcare settings [40].

A retrospective study compared phlebotomy and RBC-apheresis. Patients with PV were treated either with RBC-apheresis (n = 20) or phlebotomy (n = 20). In the RBC-apheresis group, the average RBC collection time was 25.7 ± 4.5 minutes, with a mean volume of 773.5 ± 129.3 mL. RBC-apheresis resulted in significant reductions in RBC count (0.6 × 10^12^/L [7.6%]), hemoglobin (31.1 g/L [14.8%]), and hematocrit (13.1% [20.2%]), with all changes being statistically significant (*p* < 0.001), with a marked improvement in symptoms associated [41]. Another study compared these procedures in 40 patients with PV over a four-year period. Both treatments aimed to reduce blood volume and red cell mass. Apheresis was more effective in lowering hematocrit, hemoglobin, and RBC count, with faster symptom relief and longer intervals between treatments (4–7 months vs. 20 days–4 months). Although apheresis is more expensive per session, it resulted in fewer procedures, less work absenteeism, and lower overall annual cost per patient (EUR 1800 vs. EUR 2900). Mild hypocalcemia occurred in 2.76% of apheresis cases, but this side effect was manageable. These findings suggest that apheresis, although more expensive upfront, may be a more efficient and patient-friendly alternative to conventional phlebotomy [42].

Clinical guidelines recommend RBC-apheresis as an alternative to standard phlebotomy in specific high-risk situations. These include patients with severe vascular complications where rapid hematocrit control is critical, or before emergency surgical procedures in individuals with markedly elevated hematocrit, to reduce perioperative vascular risk [43].

## 3. Therapeutic Phlebotomy in Myeloproliferative Neoplasms

PV is driven by the JAK2V617F mutation in over 95% of cases, with a minority carrying alternative activating mutations in exon 12 of JAK2 [44]. These molecular alterations lead to constitutive activation of the JAK-STAT signaling pathway, resulting in cytokine-independent hematopoiesis and an increased risk of thrombosis, hemorrhage, and disease progression. This progression may culminate in myelofibrosis (MF), where prognosis is influenced by multiple factors [45], or in acute myeloid leukemia (AML), which is associated with a poor prognosis [46,47,48].

### 3.1. Clinical Presentation and Diagnosis

PV often presents with non-specific symptoms such as fatigue, pruritus (particularly aquagenic), erythromelalgia, and early satiety due to splenomegaly. Laboratory findings include elevated hemoglobin and Hct levels, often above the diagnostic thresholds set by the WHO: hemoglobin > 16.5 g/dL or Hct >49% in men, and > 16.0 g/dL or >48% in women [9,49]. Bone marrow biopsy reveals panmyelosis, and molecular testing confirms JAK2 mutations [50].

Significantly, the clinical burden of PV is often underestimated due to its indolent presentation. However, thromboembolic events are common and may precede diagnosis in up to 25% of cases [51]. 

In fact, patients with MPNs exhibit a significantly increased risk of thrombotic complications, particularly in PV, where the incidence reaches 3.5 per 100 person-years, compared to 2.5 in essential thrombocythemia (ET) and primary myelofibrosis (PMF) [52,53].

Retrospective cohort studies, including the large-scale ECLAP study, have consistently highlighted cardiovascular events as a major cause of morbidity and mortality. In ECLAP, cardiovascular deaths accounted for 45% of all fatalities, with a thrombotic event incidence of 1.7 per 100 person-years and a cumulative incidence of 4.5% over a median follow-up of 2.8 years [54]. Non-fatal thromboses occurred in over 10% of patients. These findings are consistent with more recent data and the CYTO-PV trial, which reported post-diagnosis thrombosis rates of approximately 2.6–2.7 per 100 person-years [17,50].

The REVEAL trial demonstrated that arterial thrombosis in MPNs is primarily associated with leukocytosis, thrombocytosis, poor hematocrit control, and older age, while venous thrombosis is linked to female sex, prior thrombotic events, and leukocytosis. In polycythemia vera, venous thrombosis occurs independently of age and is associated with a high neutrophil-to-lymphocyte ratio and elevated JAK2V617F allele burden, suggesting a significant role of clonal and inflammatory factors [55].

Overall, cardiovascular complications in MPNs result from a multifactorial process involving chronic inflammation, endothelial dysfunction, oxidative stress, and elevated blood cell counts. Constitutive JAK-STAT activation drives a pro-inflammatory state with increased cytokines (e.g., IL-6, TNF-α), promoting vascular damage and thrombosis [56,57]. Endothelial activation facilitates leukocyte, platelet, and red cell adhesion, while oxidative stress and mediators like TGF-β and LOX contribute to vascular remodeling and atherosclerosis [58,59]. Neutrophils release NETs; platelets are hyperreactive and aspirin-resistant; erythrocytes show altered morphology, all enhancing thrombin generation [60].

Finally, yet central in the holistic vision of the patients, traditional cardiovascular risk factors (hypertension, diabetes, obesity, dyslipidemia, smoking) further build on the thrombotic risk, though not yet fully integrated into MPN-specific prognostic models [61,62,63,64,65]. Thus, comprehensive risk assessment should combine clinical, molecular, and lifestyle factors for optimal management.

### 3.2. Risk Stratification and Treatment Goals

The management of PV is primarily based on stratifying patients according to their risk of thrombosis. Two major categories are recognized: low-risk patients are defined as those under 60 years of age with no prior history of thrombotic events, whereas high-risk patients are those aged 60 years or older and/or with a documented history of thrombosis [66].

Irrespective of the assigned risk category, all patients with PV should be treated with low-dose aspirin, provided there are no contraindications such as bleeding disorders or aspirin intolerance. Additionally, maintaining low hematocrit levels is essential, with a therapeutic target validated in clinical trials [9]. Notably, the CYTO-PV study [17] demonstrated that strict control of hematocrit below this threshold significantly reduces the incidence of thrombotic complications. Therefore, both cytoreductive strategies and antiplatelet therapy play a crucial role in preventing cardiovascular events in PV, forming the cornerstone of modern management.

### 3.3. Treatments in PV

All patients are treated with low-dose aspirin. Early studies, such as those by the Polycythemia Vera Study Group, reported increased gastrointestinal bleeding with high-dose aspirin (900 mg daily), raising safety concerns [67]. However, the ECLAP trial in 2004 demonstrated that low-dose aspirin (100 mg daily) significantly reduced cardiovascular mortality and nonfatal thrombotic events in PV patients (HR 0.4; *p* = 0.02), without a significant increase in major bleeding risk [68].

For patients intolerant to aspirin due to conditions like gastritis, ulcers, or bleeding, clopidogrel may be considered, although data on its efficacy in PV remain limited. Its use becomes particularly relevant in cases of acute coronary syndrome, where clopidogrel is often indicated, either alone or in combination with aspirin [69,70,71].

For low-risk patients, therapeutic phlebotomy is the first-line PV and is often continued in high-risk patients as adjunctive therapy. As mentioned, this procedure involves the removal of approximately 450 mL of blood, typically 1–2 times per month during the induction phase, followed by maintenance phlebotomies based on Hct monitoring. The goal is to maintain Hct persistently <45%, a threshold established by the CYTO-PV trial as the optimal cutoff to reduce the risk of thrombosis and cardiovascular events. Here, after 31 months of follow-up, thrombotic events were significantly lower in the low-hematocrit group (2.7%) compared to the high-hematocrit group (9.8%), with a hazard ratio of 3.91 (*p* = 0.007). Including superficial vein thrombosis, rates were 4.4% vs. 10.9%, respectively (*p* = 0.02). Progression to myelofibrosis, myelodysplasia, or leukemia and bleeding events were rare and similar between groups [17].

For high-risk patients, cytoreductive treatment is mandatory. Hydroxycarbamide (HC) has long been the first-line therapy for high-risk PV, effectively reducing thrombotic events by inhibiting DNA synthesis. Early studies, including those by the PVSG and ECLAP, demonstrated its superiority over phlebotomy alone in lowering thrombotic risk and maintaining hematocrit below 45%. However, long-term use is associated with various adverse effects such as myelosuppression, gastrointestinal symptoms, reproductive toxicity, and skin complications, including ulcers and a possible risk of non-melanoma skin cancers [72,73,74].

Resistance or intolerance to HC affects approximately 10–15% of patients and is associated with increased mortality and risk of disease progression to MF or AML. The European LeukemiaNet (ELN) has established criteria for HC resistance/intolerance, though their application in clinical practice remains challenging [75,76].

Interferon-α (IFN), particularly in its pegylated and monopegylated forms (peg-IFN and ropeg-IFN), has emerged as a promising alternative, especially for younger patients and those intolerant to HC [77]. IFN has shown non-inferior efficacy to HC in achieving complete hematologic responses and offers additional benefits such as molecular remissions and JAK2 V617F allele burden reduction. The PROUD-PV and CONTINUATION-PV trials confirmed its long-term effectiveness and suggested lower rates of disease progression. A complete hematologic response (CHR), defined as hematocrit below 45% for at least three months without phlebotomy, along with platelet counts under 400 × 10^9^/L and white blood cell counts below 10 × 10^9^/L at 12 months, was achieved in 43.1% of patients (53 out of 123) treated with ropeginterferon, thereby confirming its non-inferiority compared to hydroxycarbamide. Commonly, IFN is limited by its side effect profile, including flu-like symptoms, fatigue, neuropsychiatric issues, and autoimmune disorders, which can lead to discontinuation. It is generally preferred in younger patients, especially those aiming for long-term disease modification or reduction in JAK2 allele burden, and in individuals with fewer comorbidities [78,79].

Close monitoring and dose adjustments are essential, and no universally accepted criteria currently define IFN resistance or intolerance.

Finally, JAK inhibitors (JAKi) have revolutionized the approach to MPNs [80]. Ruxolitinib (the first-in-class JAKi) is currently the only approved JAKi for PV and is indicated for patients who are resistant or intolerant to HC. The phase 3 RESPONSE trial showed that 20.9% of ruxolitinib-treated patients achieved both Hct control and ≥35% spleen volume reduction, compared to just 0.9% in the control group (*p <* 0.001). Additionally, 49% of patients experienced a ≥50% reduction in symptom burden, and 24% achieved CHR [81].

In the RESPONSE-2 trial, which included PV patients without splenomegaly, Hct control was achieved in 62% of ruxolitinib-treated patients versus 19% with best available therapy (*p <* 0.0001) [82]. The phase 2 MAJIC-PV study further confirmed its safety and efficacy, reporting CHR in 43% of ruxolitinib patients versus 26% in the control arm, and a reduction in thromboembolic events (HR 0.56, *p =* 0.05). A molecular response defined as >50% reduction in JAK2 V617F allele burden was also more common with ruxolitinib and correlated with better survival outcomes [83].

Common adverse effects include anemia, thrombocytopenia, increased infection risk (due to non-selective JAK inhibition), impaired vaccine response, and a higher incidence of non-melanoma skin cancers (OR 3.87) [84,85,86,87,88,89,90]. Some metabolic effects, like weight gain and cholesterol increases, have also been noted in other populations [91].

Despite these risks, long-term data from MAJIC-PV showed no infection-related deaths and did not confirm significant metabolic issues, suggesting an overall favorable safety profile. Barriers to routine use include cost, challenges in identifying HC resistance, and lack of guidelines for IFN failure. 

Research is ongoing, with trials such as MITHRIDATE evaluating ruxolitinib as a first-line therapy compared to HC or IFN, with a focus on event-free survival, symptom control, and molecular response (NCT04116502).

Resistance is typically functional, often due to loss of response over time, clonal evolution, or persistent inflammatory signaling [92].

### 3.4. Myelofibrosis and Post-PV Evolution

Approximately 10–20% of patients with PV will develop post-PV MF, characterized by progressive marrow fibrosis, cytopenias, and increasing splenomegaly. While phlebotomy is rarely used in advanced MF, it may still be employed in the early post-PV phase if erythrocytosis persists. However, anemia and transfusion dependence in late-stage MF often preclude further use of venesection [93]. In these cases, the therapeutic focus shifts to symptom control and disease modification through cytoreductive therapy [94,95,96,97,98].

### 3.5. Phlebotomy Limitations in Real-World Practice

Despite its established effectiveness, phlebotomy presents several practical and physiological limitations. Hematocrit measurements provide only a snapshot in time, and patients may exceed the target threshold between assessments, thereby sustaining a residual thrombotic risk [99]. Repeated procedures can also lead to iron-deficient erythropoiesis, which may paradoxically result in thrombocytosis and potentially increase the risk of thrombosis. Furthermore, the need for frequent hospital visits, venous access, and the management of related side effects can be burdensome for patients, particularly those who are elderly or have multiple comorbidities, negatively impacting treatment adherence and overall quality of life.

## 4. Future Directions in the Management of PV

While therapeutic phlebotomy remains a foundational component in the treatment of PV, growing recognition of its physiological and practical limitations has prompted exploration of alternative therapeutic strategies. As mentioned before, chief among these limitations is the episodic nature of phlebotomy, which often fails to ensure consistent Hct control. Patients may spend prolonged periods with Hct levels exceeding 45% between venesections, thus remaining exposed to thrombotic risk despite apparent control at the time of laboratory assessment.

Emerging therapies seek to overcome these limitations by providing sustained suppression of erythrocytosis, minimizing the need for phlebotomy, and, in some cases, offering disease-modifying potential.

### 4.1. Iron-Modulating Therapies and Hepcidin Mimetics

Iron restriction remains a central therapeutic principle in PV. Building on the physiological rationale of phlebotomy, several investigational agents target iron homeostasis to achieve “chemical phlebotomy”.

Rusfertide (PTG-300) is a synthetic hepcidin mimetic that reduces iron availability by downregulating ferroportin-mediated iron export. Hepcidin mimetic agents modulate erythropoiesis in PV by antagonizing the erythroferrone–hepcidin axis, thereby restricting iron availability for erythroid precursors and suppressing excessive erythropoiesis. In PV, constitutive JAK2 activation drives erythroid hyperplasia, which increases erythroferrone (ERFE) secretion from erythroblasts. ERFE acts on hepatocytes to suppress endogenous hepcidin production, resulting in increased ferroportin-mediated iron export and enhanced iron availability for erythropoiesis, perpetuating erythrocytosis [100,101]. In the Phase 2 REVIVE study (NCT04057040), rusfertide significantly reduced phlebotomy requirements, with >80% of patients achieving phlebotomy independence. Furthermore, treatment was associated with sustained Hct control and meaningful reductions in PV-related symptoms, including fatigue and pruritus [31]. Rusfertide is currently being evaluated in phase 3 trials and represents a promising non-cytotoxic approach to Hct management. Common side effects included injection site reactions, fatigue, and pruritus. Grade 3 adverse events occurred in about a quarter of patients, but no Grade 4 or 5 events were reported. One post-treatment case of acute myeloid leukemia occurred. After over 150 patient-years of exposure, 11 malignancies (mostly skin cancers) were reported, though no direct link to rusfertide was found. Thrombotic events were infrequent and occurred only in high-risk patients [102].

TMPRSS6 (matriptase-2) inhibitors represent another class of agents under investigation. TMPRSS6 is a negative regulator of hepcidin expression; its inhibition leads to endogenous hepcidin upregulation and subsequent iron restriction [103]. Both antisense oligonucleotides (e.g., IONIS-TMPRSS6Rx) and small interfering RNAs (e.g., divesiran) targeting TMPRSS6 have shown efficacy in preclinical models of β-thalassemia and PV, offering a durable means of iron suppression and erythropoiesis control. In healthy individuals, DISC-3405, a novel humanized monoclonal antibody targeting TMPRSS6, was well tolerated with no serious or high-grade adverse events (NCT06050915). Subcutaneous administration showed a favorable pharmacokinetic profile with dose-dependent increases in hepcidin and sustained reductions in serum iron, supporting monthly dosing [104]. A phase 2 trial will be recruiting shortly (NCT06985147).

Additional iron-targeting strategies include anti-hemojuvelin (HJV) antibodies, which modulate the BMP-SMAD-hepcidin signaling axis [105]. Though still in early stages of development and primarily studied in β-thalassemia and anemia of inflammation, these agents might have future applicability in erythrocytosis disorders characterized by ineffective erythropoiesis [106].

### 4.2. Epigenetic and Molecular Therapies

Targeting the aberrant transcriptional and epigenetic landscape of PV has also garnered interest. Bomedemstat (IMG-7289) is an inhibitor of lysine-specific demethylase 1 (LSD1), an epigenetic regulator involved in hematopoietic stem cell self-renewal and myeloid differentiation [107]. In a phase 2 clinical trial (NCT04262141), bomedemstat achieved durable Hct control without the need for phlebotomy in approximately 65% of treated patients at 24 weeks, while also improving symptom burden and maintaining molecular stability without progression to acute leukemia. By targeting disease biology upstream of erythrocytosis, LSD1 inhibition offers a potential disease-modifying strategy that is independent of JAK2 mutational status [108].

### 4.3. Givinostat: Epigenetic Targeting via Histone Deacetylase Inhibition

Givinostat is a selective inhibitor of class I and II histone deacetylases (HDACs) that exerts anti-proliferative, anti-inflammatory, and pro-apoptotic effects in myeloid malignancies, including PV [109]. HDAC inhibitors modulate chromatin structure and gene expression, thereby interfering with aberrant epigenetic programs that sustain malignant hematopoiesis [110].

Givinostat has demonstrated preferential cytotoxicity toward JAK2V617F-mutant cells in vitro and in vivo, with minimal effects on normal hematopoietic progenitors. This selective activity is mediated through transcriptional repression of genes involved in cell cycle progression, cytokine signaling, and inflammatory responses [111,112].

In a phase II clinical trial, givinostat showed significant hematologic and symptomatic improvement in PV patients, including reductions in pruritus, splenomegaly, and hematocrit levels. Notably, over 80% of patients achieved control of hematocrit without requiring phlebotomy during the study period. Moreover, the treatment was associated with a decrease in JAK2V617F allele burden, suggesting potential disease-modifying activity. Common adverse events included gastrointestinal symptoms, elevated liver enzymes, and mild thrombocytopenia, but the overall safety profile was acceptable [113]. These findings have prompted ongoing phase III investigations comparing givinostat to hydroxyurea in patients with high-risk PV intolerant or resistant to first-line therapies (NCT06093672).

By targeting epigenetic dysregulation while sparing normal hematopoiesis, givinostat offers a promising therapeutic option that may provide both symptom relief and long-term disease control. Its oral formulation, tolerability, and ability to reduce phlebotomy dependence position it as a viable alternative in the evolving landscape of PV.

### 4.4. Comparative Perspective

Compared to phlebotomy, these novel agents offer several potential advantages, such as stable and continuous hematocrit control, reduced need for iron depletion, improved symptom burden and quality of life, and the potential for disease modification, particularly with epigenetic and interferon-based therapies. However, challenges remain, including issues related to cost, long-term safety, and accessibility. Additionally, the optimal sequencing or combination of these agents with existing cytoreductive treatments remains to be further investigated.

A summary of the new perspectives is reported in Table 1.

## 5. Key Takeaways for Phlebotomy in PV from Current Expert Consensus


Phlebotomy is a first-line treatment in PV, aiming to reduce hematocrit and lower thrombotic risk driven by hyperviscosity. Unlike in other conditions, there are no contraindications to phlebotomy in PV due to the increased red cell and plasma volume [114]. The procedure reduces red blood cell mass, induces iron deficiency to limit erythropoiesis, and targets an Hct level ≤45%, applying both to patients with the JAK2V617F mutation and those with the exon 12 mutation, as both mutations activate the JAK-STAT pathway and lead to clonal erythrocytosis.

Some patients continue to experience hyperviscosity symptoms at 45% threshold, such as erythromelalgia or visual disturbances, and may benefit from a lower hematocrit target (40–42%). In pregnancy, the physiological drop in hemoglobin suggests a lower target may be appropriate, but supporting data are limited [61]. In high-risk pregnancies, combining phlebotomy with interferon may help avoid severe iron deficiency. Standard induction protocols recommend removing 300–450 mL of blood every other day or twice weekly until the target is reached [115,116,117]. Maintenance intervals depend on monthly hematocrit checks, later spaced to every two months. Volume replacement is not required, but hydration and blood pressure monitoring are advised. Some sources mention sex-specific normal hematocrit values (e.g., <42% for women), but these have not been rigorously studied as treatment targets in PV, and current guidelines do not recommend them. Thus, the evidence and consensus support a hematocrit target of less than 45% for all patients with PV [118,119].

Iron deficiency is an expected outcome, though it can cause symptoms such as pica or restless legs. Supplementation may be necessary in symptomatic cases, with ferritin and transferrin saturation guiding management [120]. If iron repletion increases hematocrit, cytoreductive therapy can be added.

Post-phlebotomy thrombocytosis is defined as a sustained platelet rise > 300 ×  10^9^/L without leukocytosis or splenomegaly, though its clinical significance remains unclear [121].

Phlebotomy intolerance, including repeated syncope or blood phobia, can justify cytoreductive treatment even in low-risk patients. The concept of “phlebotomy resistance,” defined as ≥ 3 procedures/year, is controversial; in most cases, higher phlebotomy needs are transient and not an indication for therapy change [122].

Furthermore, standard phlebotomy relies on intermittent Hct measurements. Since Hct levels fluctuate between procedures, patients may spend prolonged periods with elevated Hct that go undetected, thereby maintaining a residual thrombotic risk [99]. This has important implications for treatment precision and supports the need for therapies that provide more continuous control.

Figure 2 summarizes the Hct target and the most common adverse events.

## 6. Conclusions

Therapeutic phlebotomy has represented the cornerstone of erythrocytosis management for centuries, with modern applications in both iron overload syndromes and MPNs, particularly PV. Future research should focus on optimizing individualized treatment strategies that balance the benefits of hematocrit control with quality-of-life considerations and disease modification. An improved understanding of time-dependent Hct exposure, iron metabolism, and molecular disease dynamics will be key to guiding next-generation therapies in PV and beyond.

## Figures and Tables

**Figure 1 pharmaceuticals-18-01212-f001:**
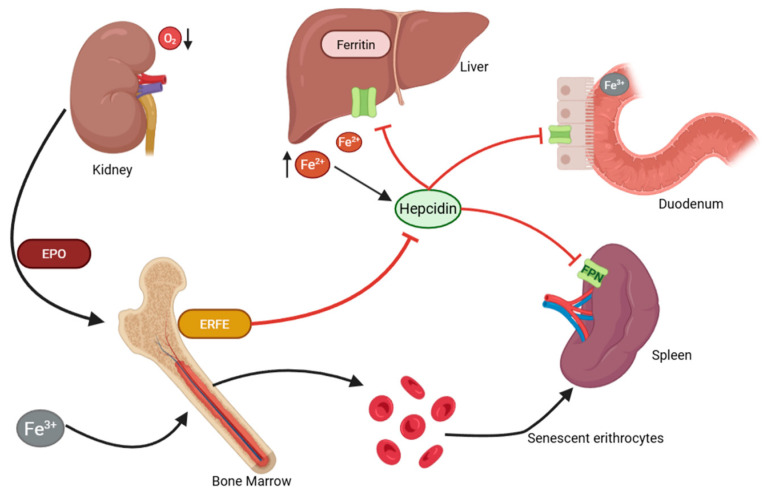
Graphical regulations of iron level and erythropoiesis.

**Figure 2 pharmaceuticals-18-01212-f002:**
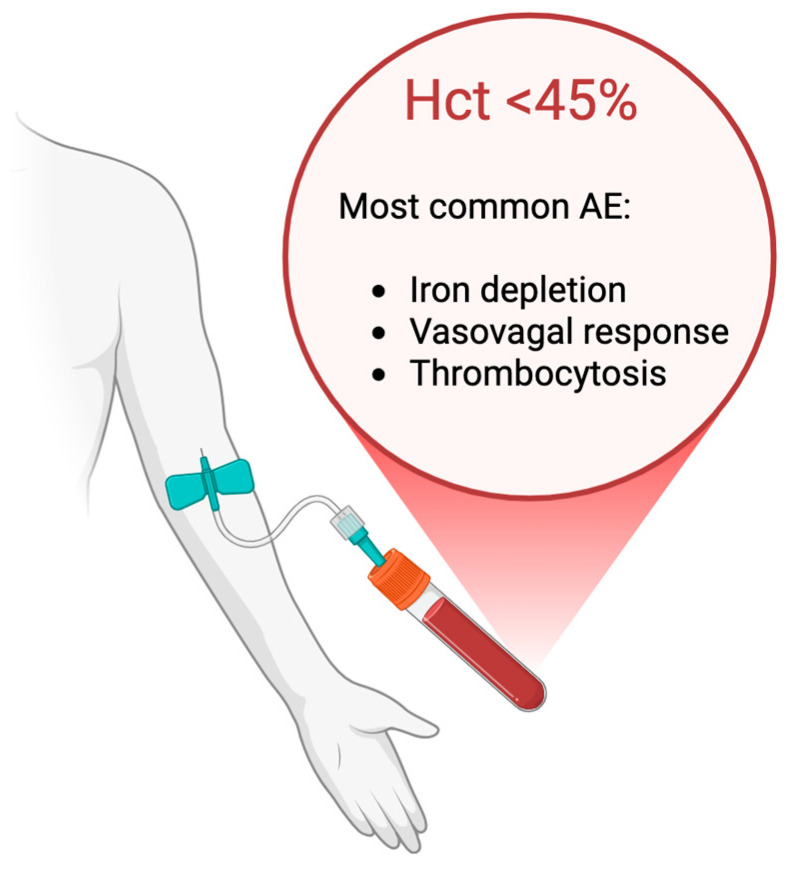
Target of phlebotomy in polycythemia vera and the most common adverse effects.

**Table 1 pharmaceuticals-18-01212-t001:** Drugs in development for the management of polycythemia vera.

Agents	Description	Clinical Results	Side Effects
Rusfertide	Synthetic hepcidin mimetic that reduces iron export	Significant reduction in phlebotomy requirements, with over 80% of patients achieving phlebotomy independence	Injection site reactions, fatigue, pruritus; grade 3 adverse events in about a quarter of patients
TMPRSS6 Inhibitors	Inhibitors of TMPRSS6 that increase hepcidin expression	Effective in preclinical models of β-thalassemia and PV	No serious adverse events in participants; favorable pharmacokinetic profile with monthly dosing
Anti-Hemojuvelin (HJV) Antibodies	Modulate BMP-SMAD-hepcidin signaling axis	Early-stage development; primarily studied in β-thalassemia and anemia of inflammation	-
Bomedemstat (IMG-7289)	Inhibitor of lysine-specific demethylase 1 (LSD1) involved in hematopoietic stem cell self-renewal and myeloid differentiation	Achieved durable Hct control without phlebotomy in about 65% of patients at 24 weeks; improved symptom burden, maintained molecular stability	No progression to acute leukemia; no significant adverse effects reported
Givinostat	Selective inhibitor of class I and II histone deacetylases (HDACs) with anti-proliferative, anti-inflammatory, and pro-apoptotic effects	Significant hematologic and symptomatic improvement; over 80% of patients achieved Hct control without phlebotomy with complete or partial response rates of 50–55% and good control of pruritus	Gastrointestinal symptoms, elevated liver enzymes, mild thrombocytopenia

## Data Availability

Not applicable.

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
