# Peer review of "Old Therapy, New Questions: Rethinking Phlebotomy in a Pharmacologic Landscape"

_pharmaceuticals, 2025, doi:10.3390/ph18081212_

Round 1
Reviewer 1 Report
Comments and Suggestions for Authors
As per the attached file

Reviewer 2 Report
Comments and Suggestions for Authors
1. The manuscript discusses maintaining hematocrit below 45% in PV but does not elaborate on the evidence or risk profile for using lower thresholds (e.g., 42%). Include supporting data or trials justifying such clinical decisions.
2. The section comparing RBC-apheresis and phlebotomy mentions cost-benefit analysis but lacks detail on outcome metrics. Recommend including statistical comparisons (e.g., effect size, confidence intervals) for Hct control duration, quality of life, and adverse events.
3. The concept of “chemical phlebotomy” via rusfertide is compelling but needs deeper mechanistic discussion on how hepcidin mimetics modulate erythropoiesis signaling pathways (e.g., erythroferrone-hepcidin axis) in PV.
4. While the paper discusses JAK2 mutations, it should better differentiate therapeutic strategies for JAK2V617F vs. exon 12 mutations, given their potential differences in erythropoietic drive and therapeutic response.
5. The review presents ropeginterferon and ruxolitinib favorably, but the adverse effect profiles, resistance mechanisms, and patient selection criteria are underexplored. Suggest adding data from registries or real-world evidence on long-term tolerability.
6. Table 1 lacks quantitative comparisons (e.g., Hct reduction %, time to response). Add a column summarizing clinical efficacy endpoints across agents to allow clearer comparison, especially in terms of replacing phlebotomy.
Reviewer 3 Report
Comments and Suggestions for Authors
This review paper outlines the current usage of phlebotomy in the treatment of Hct and PV. This work not only focused on phlebotomy itself as a treatment, namely its usage, effectiveness, side effect, and limitations, but also discussed the cause of diseases, other alternative treatments, and future aspects. This review is well organized and written in a way that it is easily understood by readers. My only suggestions is more figures and tables can be used to better introduce complicated concepts, such as the regulations of iron level and erythropoiesis. The side effect of each treatment can also be compared in a table format.
line 113-118; how the erythropoietic signal overrides iron-restrictive signals and lead to over production of red blood cell regardless iron deficiency needs more explanations. A figure can be helpful here.
line 240; "this threshold" should be specifically given.
Round 2
Reviewer 2 Report
Comments and Suggestions for Authors
Accepted!